# Comparison of the Calcium Carbide Method and Darr Drying to Quantify the Amount of Chemically Bound Water in Early Age Concrete

**DOI:** 10.3390/ma15238422

**Published:** 2022-11-26

**Authors:** Christoph Strangfeld, Tim Klewe

**Affiliations:** Bundesanstalt für Materialforschung und -prüfung, Department 8: Non-Destructive Testing, Unter den Eichen 87, 12205 Berlin, Germany

**Keywords:** concrete, hydration, material moisture, calcium carbide method, bound water, Darr method, oven drying, chemisorption, physisorption

## Abstract

Hydration is the exothermic reaction between anhydrous cement and water, which forms the solid cement matrix of concrete. Being able to evaluate the hydration is of high interest for the use of both conventional and more climate-friendly building materials. The experimental monitoring is based on temperature or moisture measurements. The first needs adiabatic conditions, which can only be achieved in laboratory. The latter is often measured comparing the weight of the material sample before and after oven drying, which is time-consuming. This study investigates the moisture content of two cement-based and two calcium sulphate based mixtures for the first 90 days by using the calcium carbide method and oven drying at 40 °C and 105 °C (Darr method). Thereby, the amount of chemically bound water is determined to derive the degree of hydration. The calcium carbide measurements highly coincide with oven drying at 40 °C. The calcium carbide method is therefore evaluated as a suitable alternative to the time-consuming Darr drying. The prompt results are seen as a remarkable advantage and can be obtained easily in laboratory as well as in the field.

## 1. Motivation

In 2020, the global human-made mass exceeded all living dry biomass [1] and has been further increasing exponentially [2]. More than 95% of this anthropogenic mass consists of hydrophilic porous materials and the biggest share is concrete with around 40% [1]. The cement industry is responsible for around 8% of the global CO${}_{2}$ emission [3], which is estimated to grow by 12–23% until 2050 [4]. However, concrete does not last forever. High investments in the maintenance of modern transport infrastructure are required. For example, around 13 billion euros are needed in Germany every year [5]. Furthermore, new structures have to sustain much higher loads due to an almost permanent increase of freight transport during the last decades [6]. This is also reflected in an increase of assumed traffic loads to be considered for the design and evaluation of the load-bearing capacity of new bridges [7]. Therefore, the application of specialised concrete mixtures such as ultrahigh-performance concrete or high-strength concrete comes into question. Nevertheless, these maintenance and building activities generate negative impacts on the environment as well. Per tonne of cement, around 580 kg of CO${}_{2}$ are emitted, mainly due to the chemical process of decarbonation [8,9]. Thus, as a substitute for cement, “alternative clinkers”, which have a lower chemical CO${}_{2}$ emission, are promoted nowadays [8,10,11]. Furthermore, the demolition of existing structures generates huge amounts of waste. For example, 36.4% of the overall waste (by weight) of the EU comes from the construction sector [12]. This clearly shows the saving potentials if recycled building materials are used. Modern concrete structures should sustain higher loads with a simultaneous reduction of CO${}_{2}$ and the usage of recycled construction material.

New binders, new mixtures, the use of recycled materials or aggregates all represent very promising approaches. Creating confidence in these new materials requires reliable in situ characterisation. In construction, one crucial parameter is the degree of hydration (DoH). For instance, based on this parameter, the current and final compressive strength are estimated, or concrete post-treatment actions are scheduled and optimised. Therefore, this study discusses an approach for quantifying the DoH which works in the field as well.

## 2. Theory of Chemisorption of Water in Early Age Mixtures

Although concrete is designed to last for several decades, its essential properties are determined during the first hours, days and weeks after concreting. As soon as the dry bulk material comes into contact with mixing water, the hydration process starts immediately [13]. During this chemical reaction, the resulting microstructure is developed. Within this process, the liquid mixing water undergoes a change of state. A significant part of it is hydrated, i.e., the liquid water is absorbed by the solid matrix and remains chemically bounded [14]. In addition, physisorption occurs, i.e., liquid water is adsorbed at the fringe of the emitting pore network [15]. Although this water interacts with the surface of the solid cement matrix, its binding energy is much lower compared to chemically bound water, even in the case of monolayer adsorption [16]. Thus, in distinction to chemically bound water, adsorbed liquid water is also called physically bound water. In most cases, more mixing water is present in the pore network as it would be required for absorption and adsorption. This water is transported to the boundary of the solid structure by hydraulic conductivity or diffusion within the pore network and evaporates at the outer surface [17]. This part of the mixing water is also called evaporable, mobile, or free water. Nonetheless, the absorption and adsorption of mixing water are always closely linked [18]. The hydration generates a fine pore structure with an increasing inner surface at which physisorption takes place [19]. These two processes are jointly responsible for the self-desiccation of concrete [18]. If there is not enough liquid water available, the hydration process is unintentionally decelerated or even stops entirely [20]. This has probably strong negative effects on concrete strength, durability and other properties, leading to a significantly reduced lifetime of the entire structure. A precise modelling and quantification of the entire hydration process at an early state is important to ensure a high level of long-term concrete performance [13,21].

Figure 1 shows a schematic sketch of the evolution of water and its different states of matter during the hydration process. In the beginning, the dry bulk material in solid state and the mixing water in liquid state are separated. In this sketch, all quantities are given in weight. The sum of the initial weight of dry bulk material and mixing water gives the maximum reachable weight of the hydrated sample. During concreting, both are mixed together, and the hydration starts immediately [13]. In the considered scenario, evaporation is taken into account as well. The mixing water takes all three states of matter. A large portion of the mixing water is used for the hydration of the bulk material. In this process, several hydration products are formed and the mixing water is bound into the material matrix [22,23,24]. Although all bound water could be re-extracted at very high temperatures of more than 1000 °C [25], at normal ambient conditions the water becomes part of the solid phase and remains there. This leads to an increase of the dry sample weight over time, because the hydrated water is assigned to the dry material matrix. A portion of the mixing water evaporates at the outer surface of the sample. This water in gaseous phase is adsorbed by the ambient air and does not contribute to the hydration process any more. Hence, the total weight of the moist sample decreases over time. Both hydration and evaporation lead to a reduction of the mixing water in liquid phase and this process continues until a thermodynamic equilibrium is reached. At this state, the dry material weight remains constant because the hydration process is finished. Furthermore, no evaporation occurs any more, because the humidity level within the pore network of the sample is in equilibrium with the ambient conditions as well. However, even at this stage, water in liquid phase is still present in the system. This is the physisorption of water at the inner surface of the pore network and can be determined, following various adsorption theories [26,27,28]. This remaining part of the mixing water is the material moisture. The adsorbed liquid water in the pores contains dissolved salt and hydroxide ions, leading to different chemical properties compared to the initial mixing water from the tap. Although the thickness of the adsorbed water layer is in the lower nanometre range [27,29], it is still in liquid phase. The upper right area of Figure 1 shows the final amount of water in each of the phases: the evaporated water in gaseous phase, the adsorbed water in liquid phase and the absorbed/hydrated water as part of the solid phase.

Finally, the ratio between the amount of chemically bound water divided by the ultimate amount of chemically bound water at thermodynamic equilibrium represents the current state of hydration [30,31]. At thermodynamic equilibrium, the dry sample weight is supposed to be constant. In case of altering the material itself, e.g., by carbonation, the dry sample weight might alter as well. The older the sample, the stronger the possible deviations. In our experiment, all samples were freshly made, so the influence of carbonation was negligible. Thus, a monitoring of the hydration process was possible by measuring the dry weight of the material matrix over time. As already mentioned, there are several techniques and settings for material drying. To overcome this uncertainty of choosing the “right” method of drying, a second approach is to measure the amount of water in liquid phase and subtract this from the total weight of the moist sample to yield the dry sample weight. Following the second approach, experimental CCM measurements were carried out in this study.

## 3. State of the Art for Measuring the Degree of Hydration

The hydration process itself includes the formation of several hydration products such as calcium silicate hydrates (C-S-H), ettringite, monosulphate, calcium hydroxide, etc., [22,23,24]. In general, two parameters are used to quantify this complex process; the isothermal heat of hydration *Q* or the amount of chemically bound water $w_{b}$ [30]. To derive the current state of hydration, these two parameters are set in relation to their final values $Q_{\infty }$ and $w_{b,\infty }$ when the hydration is completed. Finally, both $Q/Q_{\infty }$ or $w_{b}/w_{b,\infty }$ theoretically yield the current state of hydration [30]. Both parameters show an asymptotic approach towards the unit value but never reach this value in practice. This would require an ideal setup including the “perfect” water-to-cement ratio for full hydration and ideal hydration shells around the cement grains [14,32]. The experimental procedure for measuring *Q* based on calorimetry is widely established [18,19] and already standardised [33,34]. However, adiabatic or isothermal conditions are required, which makes this method only feasible in the laboratory. As a method was required which worked reliably in the field as well, calorimetry was not considered further in this study.

Another parameter than temperature is required to describe the process in the field. Therefore, several approaches exist to determine the degree of hydration based on $w_{b}$, i.e., the amount of bound water. In a first step, the amount of physically bound water has to be determined and/or removed. Probably the most common method is the Darr drying in an oven [35]. Choosing the right drying temperature is already a widely discussed issue with recommended temperatures between 40 °C and 105 °C for cement-based materials [36,37,38,39,40]. Ettringite and gypsum might already become unstable at temperatures above ca. 60 °C [36,41,42], as well as the interlayer water between the C-S-H sheets due to cavitation [43]. It must also be considered that the oven drying may lead to a significant bias in the data, because the hydration process is not stopped immediately after sample extraction. On the contrary, increased ambient temperatures lead to an accelerated hydration as long as sufficient moisture is available within the sample [14,30]. Oven drying provides, in fact, only single point measurements and the experimental values might not be representative of the current state of hydration. Therefore, other approaches were developed to determine the material moisture during hydration such as ground-penetrating radar [44], time-domain reflectrometry [45], electrical impedance and conductivity [46], piezoelectric sensors [47], relative humidity measurements [48,49], nuclear magnetic resonance relaxometry [50] or infrared spectroscopy combined with scanning electron microscopy [51]. Nevertheless, all these mentioned measurement techniques require a precise calibration for the investigated cement mixture at controlled ambient conditions. This high laboratory effort prevents these methods from being used in the field. An alternative approach is the use of several “sister-samples” instead of investigating the sample of interest directly. All samples are made of the same batch of mixture and exposed to identical environmental conditions. Nevertheless, the challenge of measuring the “true” amount of chemically bound water remains.

Despite the challenges in measuring the DoH, the results are highly dependent on the investigated material. Jennings showed a DoH of around 55% after three days for a cement-based material with a water-cement ratio of w/c=0.4 based on calorimetry measurements [52]. Similar to this, Di Luzio et al. measured a DoH of 60% after 7 days for an ordinary Portland cement with w/c=0.4 [31]. Cook et al. investigated mixtures of Type 1 Portland cement with 0.3≤w/c≤0.7. After 10 days, the DoH was between 45% and 55%, whereas different methods of sample drying were evaluated [53]. A Montalieu cement with w/c=0.35 was investigated by Bentz et al. [32]. There, a DoH of around 82% was already reached after 4.5 h. Scrivener et al. found that for a white cement paste with w/c=0.4, after 10 days, only around 5% of the initial capillary water was present. All these mentioned studies were carried out in the laboratory at controlled ambient conditions and with precisely characterised mixtures. Today, the use of climate-friendly cement and recycled building materials comes more into focus [8,10,11,54]. To determine the DoH of such new and often rather unknown mixtures based on the amount of chemically bound water will be very challenging, even in the laboratory. Ideally, the applied measurement technique should be applicable in the field as well.

In this study, we used the calcium carbide method (CCM) to derive the amount of chemically bound water as an indicator of the DoH. Calcium carbide reacts with the evaporable water of a sampling material in a pressure vessel. The reaction leads to a measurable pressure increase, which is proportional to the amount of free water [55]. Thereby, this method has several advantages. It is easy to use both in laboratory and in the field. The sampling material is investigated directly without any kind of sample drying. This means there is no time delay and no uncertainty about the drying procedure. Therefore, the measurement value is representative for the current state of hydration and no calibration for the material composition is needed. We investigated two cement-based and two calcium sulphate based mixtures for the first 90 days. The CCM measurement were compared with oven-dried samples at 40 °C and 105 °C for validation.

## 4. Materials

In this study, two cement-based and two calcium sulphate based materials were investigated. The dry bulk material was purchased as bagged goods from a local hardware store. The properties of the different materials are listed in Table 1. As the binder and the aggregates were already premixed, the water demand was given in litre per kilogram of dry screed material. The mixing water itself was ordinary tap water. The binder of the screed was identical to concrete, except for a maximum aggregate size of 8 mm, which was higher for concrete. For each screed type, three different sample heights *h* of 50 mm, 60 mm and 70 mm were investigated. A new sample was required for each of the nine intended measurement days for the destructive CCM testing. In total, 27 samples of each calcium sulphate screed type, and 54 samples of each cement-based screed type were produced from one respective mixture batch. The number of samples for the cement-based screeds was doubled to investigate the difference between Darr drying at 40 °C and 105 °C. Both ovens had no humidity control, they used just supply air from the ambient. The ambient conditions around the ovens in the experimental hall were estimated to be around 23 °C and 40% relative humidity.

## 5. Experimental Approach

The CCM requires a small portion of sampling material. A certain amount of material of around 10 g to 100 g was extracted and needed to be crushed into small particles. Then, the sampling material was weighed and filled in a pressure vessel with known inner volume. A certain amount of anhydrous calcium carbide was added in a glass ampulla. Steel balls were placed in the pressure vessel as well for further crushing of the sampling material. Then, the pressure vessel was sealed and the pressure recording started. By shaking the vessel by hand, the sampling material was ground and the calcium carbide ampulla cracked. At this point, the chemical reaction started [55].

The calcium carbide reacted with the evaporable water of the sampling material in the sealed pressure vessel. The reaction took place at the surface of the calcium carbide particles and had to be in contact with the sampling material (through crushing and shaking). Calcium carbide and liquid water reacted to produce solid calcium hydroxide and gaseous acetylene (CaC${}_{2}$ + 2H${}_{2}$O → Ca(OH)${}_{2}$ + C${}_{2}$H${}_{2}$). The acetylene occupied more volume than the initial materials, leading to a pressure increase within the pressure vessel. Each molar volume of water led to the same molar volume of acetylene, independent of the material composition. After the procedure was terminated and the resulting pressure was corrected according to the temperature inside the vessel, the amount of liquid water was known. At this point, the entire chemical reaction was not necessarily finished, i.e., a small amount of the material moisture had not reacted yet. Calcium carbide might react in the same way with methanol, so the sampling material must be free of methanol to avoid a systematic measurement bias [55].

According to Figure 1, the weight of the dry material could now be derived. The moist sample material was weighed before filling the sampling material into the pressure vessel. Based on the CCM, the weight of liquid water was measured. Subtracting the two weights gave the dry material weight. This included the dry bulk material and the hydrated water. Knowing the initial weight of the sister sample and its *w*/*c*-value, the weight of chemically bound water could be determined, independent of the material composition and without the uncertainties of sample drying.

The corresponding equations for the material moisture and weight are briefly summarised. Here, *m* denotes the mass in kg and *M* stands for the material moisture by weight in kg of water per kg of material. The material can be considered in two conditions, dry or moist, which leads to different reference masses and moisture values, so that there are two definitions for the material moisture, $M_{moist}=\frac{m_{H_{2}O}}{m_{moist}}$ or $M_{dry}=\frac{m_{H_{2}O}}{m_{dry}}$. Equation (Equation 1) gives the relation between the dry mass $m_{dry}$ and the moist mass $m_{moist}$. The difference between both is the physiosorbed mass of moisture $m_{H_{2}O}$.
(1)$$\begin{array}{ccc} m_{dry} & = & m_{moist}-m_{H_{2}O}=m_{moist}-m_{wet}\cdot M_{moist}=m_{moist}\cdot \left(1-M_{moist}\right) \end{array}$$

The Darr method directly yields the dry mass, and hence $M_{dry}$. Although the CCM approach measures $M_{moist}$, the underlying calibration functions are mostly given in $M_{dry}$ [55]. Equation (Equation 2) yields the calculation of $M_{moist}$ based on $M_{dry}$. This formulation is free of any measured mass and therefore generally applicable.
(2)$$\begin{array}{ccc} M_{moist} & = & \frac{1-M_{dry}}{-M_{dry}+\frac{1}{M_{dry}}} \end{array}$$

To describe the hydration process according to Figure 1, the determined masses were normalised by the initial mass $m_{initial}$, which was the summation of the dry bulk material and the mixing water. Following the initial identity of $m_{dry}=m_{moist}-m_{H_{2}O}$, the two different measurement principles were directly comparable.
(3)$$\begin{array}{ccc} \frac{m_{moist}\cdot \left(1-M_{moist,CCM}\right)}{m_{initial}} & \hat{=} & \frac{m_{dry,Darr}}{m_{initial}} \end{array}$$

## 6. Testing Procedure and Sample Preparation

To cast the samples, cylindric formworks were designed as shown in Figure 2. Therefore, sliced polyvinyl chloride pipes were attached and sealed on a film-faced sheet of plywood with silicone. A vertical slit closed by a hose clamp was applied to ensure an easy stripping of the formwork after casting. All screed mixtures were produced according to the manufacturers’ instructions and then poured into the formworks. With a vibrating table, an appropriate compression was achieved. Unlike the drawing in Figure 2, multiple formworks were attached to one single sheet of plywood to accelerate the described process of casting and compressing. Because of this, it was not possible to weigh each individual sample immediately after casting. Instead, this was postponed to the day after casting. During this time all fresh samples were covered with a PE foil to limit the evaporation of mixing water.

After a 20-h curing time, the formworks were removed, and each sample was weighed to obtain $m_{initial}$. Then, the bottom and side surfaces were covered with PE foil and waterproof tape, respectively, to allow a realistic self-desiccation from above. For the remaining testing time, all samples were stored under controlled climatic conditions of 23 °C and 50% relative humidity. The first measurement day was 7 days after casting followed by tests after 10, 14, 21, 28, 35, 50, 70 and 90 days. For CT-C25-F5, the first measurement had to be discarded due to the use of outdated calcium carbide.

A measurement day included the following procedure:The respective sister sample was removed from the climate chamber and unwrapped from its waterproof tape and PE foil cover.The sample was weighed to obtain $m_{moist,total}$ for the CCM.CCM testing was performed.(a)A small portion of material was taken off destructively from the sample:CT screed: Since there were two samples for each measurement day, 20 g was collected from the upper part of the first, and from the lower part of the second sample. The broken off piece was approximately 30 mm deep on each side, independent of the sample thickness.CA screed: according to the expected moisture content, 20 g (7 days old), 50 g (10 days or older) or 100 g (35 days or older) were collected over the entire height of the sample.(b)The CCM procedure was further conducted as described in Section 5 to obtain $M_{moist,CCM}$. According to the manufacturer’s manual the time intervals for shaking the vessel were: (start) 1 min shaking—3 min rest—1 min shaking—5 min rest—10 s shaking (end).The remaining sample was weighed to obtain $m_{moist,remain}$ for the Darr drying method.The remaining sample was then put into an oven:(a)CT screed: one sample was dried at 40 °C, one at 105 °C.(b)CA screed: dried at 40 °C.

All measurements were carried out by the same person to reduce the influence of the human factor. This was especially relevant for the CCM testing, where the intensity of shaking could have an influence on the release of reactive liquid water. To avoid this, commercial CCM manufacturer might offer stir sticks connected to an electric drill to generate a recurring crushing of the material. If flinty stones are included in the material, the stir sticks avoid any sparking which might ignite the gas inside the pressure vessel. The emitted pressure wave is not a safety issue, but would probably destroy the membrane of the integrated pressure sensor.

During the drying process, the sample weights were controlled in appropriate intervals of one to three days. The samples were removed from the oven and were left for 20 min to cool off, before being weighed. According to [35], $m_{dry}$ was reached when the difference in mass on three consecutive weighings (with at least 24 h in between) did not change more than 0.1% of the total mass. The average drying time for the CT screed was 21 to 22 days at 40 °C and 7 to 8 days at 105 °C. The CA screed dried after 6 to 7 days at 40 °C.

## 7. Results

In a first step, the results of the measured material moisture are compared between the CCM and Darr method. For the latter, the influence of the drying temperature is discussed as well. Afterwards, the amount of chemically bound water is derived.

### 7.1. Comparison of Material Moisture Based on Darr and CCM

Figure 3 shows the measured material moisture of the calcium sulphate based samples over time. For both screed types, an oven temperature of 40 °C is recommended for Darr drying. The first measurements started 7 or 10 days after concreting, showing the highest moisture values for both methods. After around 70 days, both sample moisture values were close to equilibrium. The CCM gave constant material moisture and the Darr drying still indicated a slightly decreasing moisture. Nevertheless, all four lines showed the expected asymptotic trend. Furthermore, the correlation was high between the CCM and Darr method. If unintended hydration had occurred during oven drying, the material moisture would have decreased. The opposite was measured for both samples. The Darr drying gave higher moisture values for at least the first 20 days. These measurements indicated that hydration was negligible in the time frame between sample extraction and the end of oven drying. In this case, both approaches seemed to be suitable for the measurement of material moisture, and in consequence, for the determination of chemically bound water as well.

The same analysis as described above was conducted for the cement-based sample CT-C25-F5. For cement-based material, there is no unified drying temperature [39]. For practical reasons, a temperature of 105 °C was chosen, which yielded the shortest measurement time. Nevertheless, several studies proved that this high drying temperature is already able to dissolve chemically bound water [36,37,38]. In this context, the mineral ettringite which is formed during hydration plays a role [22,23,24]. The water-rich ettringite mineral becomes unstable at temperatures above ca. 60 °C and/or at low relative humidity below 30% RH [36,41,42]. Hence, oven drying at 105 °C might cause systematic deviations regarding the material moisture. Therefore, a second oven drying temperature was chosen of 40 °C with around 15% RH. Furthermore, sampling material was extracted individually from the upper part of the first, and from the lower part of the second CT sample, as described in Section 6.

Figure 4 depicts the resulting moisture content for all four approaches for CT-C25-F5. The Darr drying was done at 40 °C or 105 °C, respectively. The drying temperature of 105 °C yielded significantly higher material moisture values. The deviation between both was between 1.6 wt.-% and 2.1 wt.-%. This clearly indicated that with the 105 °C drying temperature the amount of physiosorbed (evaporable) water could not be correctly determined. The solid cement structure was partly damaged and chemically bound water was released and evaporated. The sample weight decreased, which was incorrectly recognised as a higher material moisture. In consequence, the 105 °C data were not considered in further analyses. In addition, the two CCM measurements from the upper and lower part are shown. The upper part, which includes the top surface where the evaporation takes place, shows lower moisture values compared to the lower part. Furthermore, the moisture was significantly below the one recorded for the Darr drying at 40 °C. The main reason was the moisture gradient within the sample. Comparable measurements were carried out with cement-based screed samples with a thickness of 70 mm [17,56]. Here, it is shown that the moisture gradient could vary by more than 2 wt.-% within the sample during hydration and drying. As Darr drying included the entire sample, the CCM approach should consider the entire cross section as well. In consequence, the values of the upper and lower parts were averaged to yield a representative CCM measurement for the entire sample. This mean value was used for further analyses in this study.

In conclusion, the Darr drying at 40 °C quantified the evaporable moisture, while keeping the solid cement matrix intact. Because the entire sample was dried in the oven, obviously the entire cross section was affected by this method. In consequence, it was compared with the mean CCM of the upper and lower parts. This should be a representative value, despite a certain moisture gradient. As shown in Figure 4, the two curves coincide. Within the first 30 days, some deviations are visible with a maximum deviation of 1 wt.-% on day 10. In the beginning, the samples were very moist, and the carbide was probably not able to react with all adsorbed water within the common 10 min. An adapted measurement procedure with an extended time period would probably decrease the deviation between Darr drying and the CCM with the first 30 days.

Figure 5 presents the same analysis for the second cement-based material CT-C40-F7. All measured moisture values were lower compared to those of CT-C25-F5. This was mainly caused by the lower water demand for CT-C40-F7, see Table 1. A bigger portion of the mixing water was required for the hydration process and less evaporable water remained in the sample. Hence, the measured material moisture was reduced. The material moisture deviations between the two different Darr drying temperatures were even more prominent, if compared to CT-C25-F5, in absolute and relative numbers. For CT-C40-F7, the 105 °C drying resulted in moisture values almost twice as much as for 40 °C. This indicated that the solid material matrix was even more affected and damaged by the high drying temperature. The CCM measurements of the upper and lower parts were closer to each other, mainly caused by the overall reduced material moisture. For this material, the averaged CCM moisture values were above those of the Darr drying at 40 °C. This systematic deviation was also observed for the two other samples with 50 mm and 60 mm thickness. The authors’ hypothesis to explain the deviation was the higher amount of cement paste in the material. It was more difficult for the carbide to reach the entire adsorbed water in the pore network. Thus, not all water had reacted after 10 min.

For CT-C25-F5, the Darr moisture was above the CCM moisture within the first 40 days. In case of CT-C40-F7, the moisture was similar at the two first measurement days. Then, the Darr moisture decreased more rapidly than the CCM moisture. If unintended hydration during oven drying occurred, the trends would be opposite to each other. Therefore, the measurements indicated that hydration during Darr drying after sample extraction was negligible and did not affect the amount of chemically bound water.

Based on the material moisture measurements, the CCM showed a satisfying correlation to the Darr drying at 40 °C for both investigated materials. In a next step, the amount of chemically bound water was calculated based on the CCM measurements.

### 7.2. Determination of the Chemically Bound Water Based on Darr Drying and CCM

According to Figure 1, subtracting the physisorbed (evaporable) water from the sample weight of the moist material gives the dry material weight. This is the weight of the dry bulk material plus the hydrated water. In contrast to other approaches, this method avoids sample drying and/or a time delay between sampling material extraction and receiving a measurement value.

Figure 6 shows the results based on this approach for the two calcium sulphate based screeds. All weights were normalised by the initial weight during concreting, which was the weight of the dry bulk material plus the weight of the mixing water, according to Figure 1. In Figure 6, the normalised weight of the moist sample decreased over time, caused by the evaporation of the mixing water. Both samples tended asymptotically towards their final values when the hydration was finished and the complete sample had reached its equilibrium moisture content [56]. This final condition could probably not be reached after the presented 90 days. According to Equation (Equation 3), the depicted dry weight was determined experimentally in two ways. The first approach was based on the CCM. The dry mass was obtained by measuring the moist sample weight and the material moisture. The second approach was the common Darr drying in the oven at 40 °C. In addition, the normalised dry bulk material is shown. This would be the lower limit, if the entire mixing water evaporated and no water adsorption or adsorption took place in the bulk material. The theoretical value is a direct function of the *w*/*c* value, as shown in Equation (Equation 4). Because both materials had the same initial water demand during mixing, the normalised dry bulk material was identical for both screed types.
(4)$$\begin{array}{c} \frac{m_{bulk}}{m_{initial}}=\frac{1}{1+(w/c)} \end{array}$$

For both samples, the normalised dry weight coincided for the CCM and Darr method. Within the first 20 days, a slight scatter was observable. For the subsequent 70 days, the values were almost identical. Furthermore, for both materials, the values were nearly constant. This indicated that the hydration process was completed already before the first measurement on day seven or ten and proved that the entire solid material matrix had been formed entirely within one week. Although the initial water demand was identical for both samples, the dry material weight was approximately one percent higher for CA-C30-F7. This mixture is designed to withstand a higher compressive strength, as stated in the published data sheet, provided by the manufacturer. This is generally achieved with a higher material compaction due to an increased hydration rate. Related to the available water in the beginning, around 7.7% additional mixing water was hydrated.

Figure 7 presents the same approach for the two cement-based materials. Because these two mixtures had a different water demand, the lower limit expressed by the dry bulk material weight varied. As expected, the normalised weight of the moist samples decreased with time due to water evaporation. For the CT-C40-F7 sample, the Darr dried samples weight was above the CCM value during all measurement days. In the case of CT-C25-F5, the trend was the opposite, except for the last measurement point on day 90.

Between day 7 and 21, both methods showed that the dry sample weight increased for the CT-C40-F7. However, later, it also decreased again for both methods. These fluctuations were probably caused by the general measurement uncertainty and a clear trend could not be derived. Eventually, no significant increase in dry weight could be determined. This indicated that for the cement-based materials, the hydration was also completed within the first week. This is contradicting other studies, because the formation of C-S-H is supposed to take several months [22,23,24]. On the other hand, this formation is a reformation within the solid cement matrix, e.g., ettringite transforms to C-S-H and monosulphate [23]. This process might not need evaporable water, although it is an exothermic reaction, which is measurable with calorimetry. In such a case, monitoring the amount of chemically bound water could not disclose the hydration process entirely.

However, this study’s central subject was the experimental determination of chemically bound water. In conclusion, the CCM approach coincided well with the Darr dried sample at 40 °C. Therefore, the CCM was able to appropriately quantify the amount of chemically bound water and could be used for hydration monitoring. This approach avoids long-term oven drying and delivers contemporary results in the laboratory as well as in the field. Furthermore, by modifying the measurement procedure, this method also enables the on-site quantification of the crystalline-bonded water of ettringite incorporated in the material sample [57].

### 7.3. Comparison of Darr Drying and CCM Applied to Quantify Chemisorbed Water

Darr drying and the CCM were able to quantify the evolution of the chemisorbed water and the hydration process. In Figure 8, both methods are compared to each other based on a common linear regression. The regression includes only one coefficient, the slope. A y-axis intercept is excluded for two reasons. First, an offset between both methods is unreasonable, because a totally dry material should be quantified as zero for both methods. Second, the definition area is D=R(0,1), but the measured values are all above 0.92. Here, the sampling points have a large leverage towards the origin of coordinates. Thus, a small scatter of the data points would have an exaggerated influence on the estimated y-axis intercept.

The slope coefficient *s* was calculated based on all 102 sampling points of the normalized dry weight. Here, only the Darr drying values at 40 °C were included, which led to s=0.9995. The corresponding coefficient of determination was $R^{2}=0.958$. The slope value was very close to one and around 96% of the variance of the data could be explained by this simple linear regression model. This confirmed the high agreement between Darr drying at 40 °C and the CCM for determining the DoH. On the other hand, the Darr values at 105 °C showed a systematic negative bias of around two percentage points. Once again, this validates that such a high drying temperature is able to extract chemisorbed water from the investigated material samples. Furthermore, looking on the residuals of the regression, there was no indication that the different sample thicknesses had any influence on the determination of the DoH.

## 8. Conclusions

The essential properties of hydraulically bound materials were determined during the first hours, days and weeks after concreting. A proper hydration process is crucial for structural performance and long-lasting durability. The process can be measured and monitored by calorimetry or by determining the amount of chemically bound water. For the latter approach, the time-consuming Darr drying method is often used which has several systematic measurement uncertainties, such as the selection of the correct drying temperature. In this study, the CCM is presented and evaluated as an alternative method. This method delivers contemporary results, can be used in the field as well and covers all kinds of material compositions and binders. This makes the method highly suitable to monitor the DoH of CO${}_{2}$-reduced binders, new mixtures or recycled materials. In case of unexpected ambient conditions such as heat and drought periods, this in situ method, which delivers prompt results, enables a dense DoH monitoring if required. The main findings were:The CCM directly measured the physiosorbed (evaporable) water and hence was capable of monitoring hydration.The measured material moisture and the derived hydration curves of CCM highly coincided with Darr measurements at 40 °C oven temperature.Darr drying at 105 °C highly affected the solid material structure and dissolved chemically bound water. This systematic deviation influenced the measured material moisture by more than 50% for the cement-based materials investigated here.The quantification of chemically bound water with the Darr method and the CCM showed that no significant chemisorption of free water occurred between measurement day 7 and 90 for the calcium sulphate based and the cement-based materials.The formation of the C-S-H generally takes several months. The formation and reformation of C-S-H, ettringite, monosulphate and calcium hydroxide might represent an exothermal process, which is measurable with calorimetry. However, the current measurements indicated that this process was unaffected by the amount of evaporable or chemically bound water.

The CCM coincided well with Darr drying at 40 °C. Therefore, the CCM is an alternative to the time-consuming oven drying. Due to its easy handling and independence of the material composition, it can be applied to all scarcely researched binder types. For both established or new “alternative” materials, the monitoring of the hydration process and its experimental quantification remains crucial for the creation of safe and long-lasting structures.

## Figures and Tables

**Figure 1 materials-15-08422-f001:**
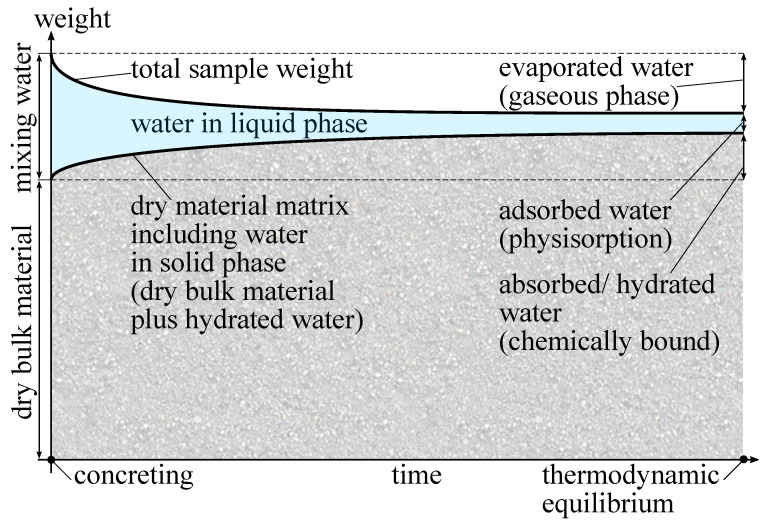
Schematic sketch of the evolution of the mixing water and its phase transition.

**Figure 2 materials-15-08422-f002:**
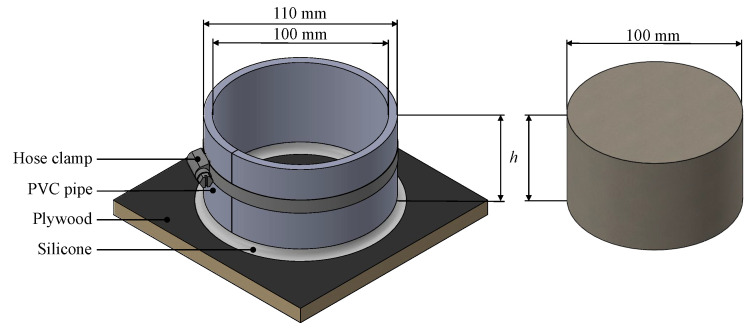
Formwork (**left**) to cast the resulting screed sister samples (**right**) of 100 mm diameter and varying heights *h* of 50 mm, 60 mm and 70 mm.

**Figure 3 materials-15-08422-f003:**
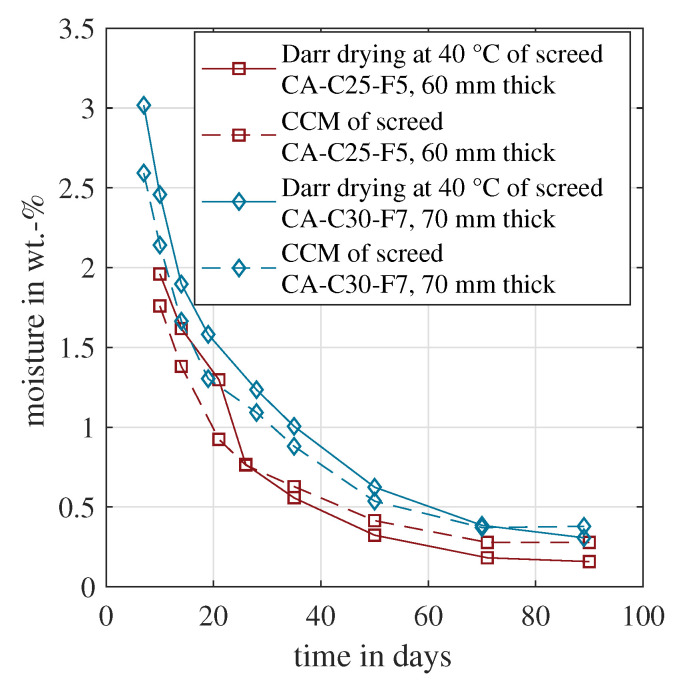
Evolution of the material moisture measured by CCM and Darr drying for the calcium-sulphate-based samples CA-C25-F5 and CA-C30-F7.

**Figure 4 materials-15-08422-f004:**
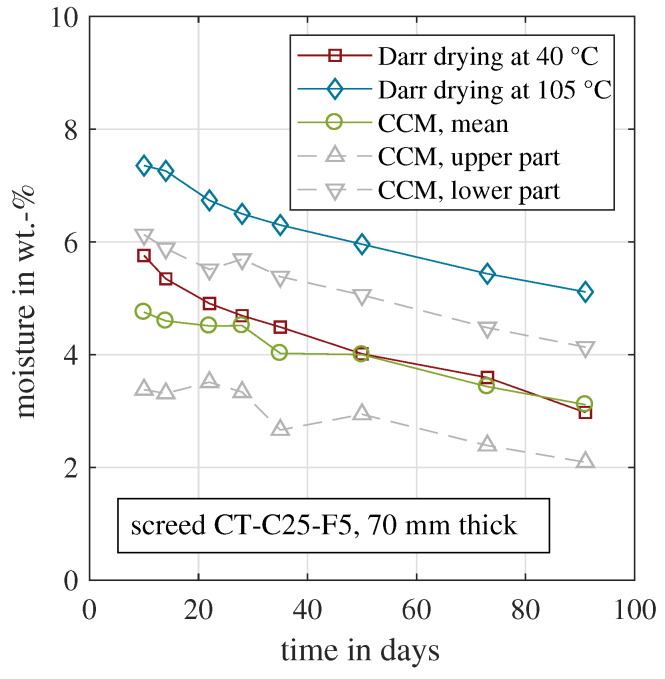
Evolution of the material moisture measured by CCM and Darr drying for the cement-based samples CT-C25-F5.

**Figure 5 materials-15-08422-f005:**
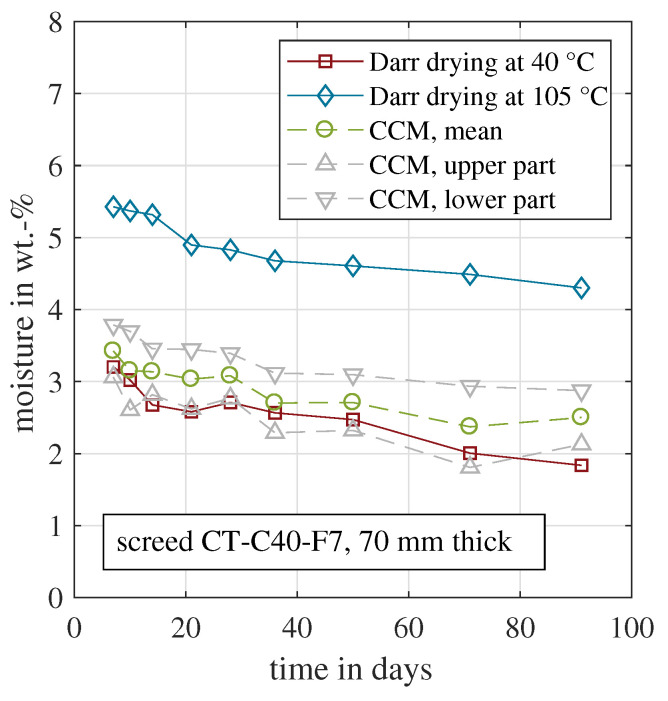
Evolution of the material moisture measured by CCM and Darr drying for the cement-based samples CT-C40-F7.

**Figure 6 materials-15-08422-f006:**
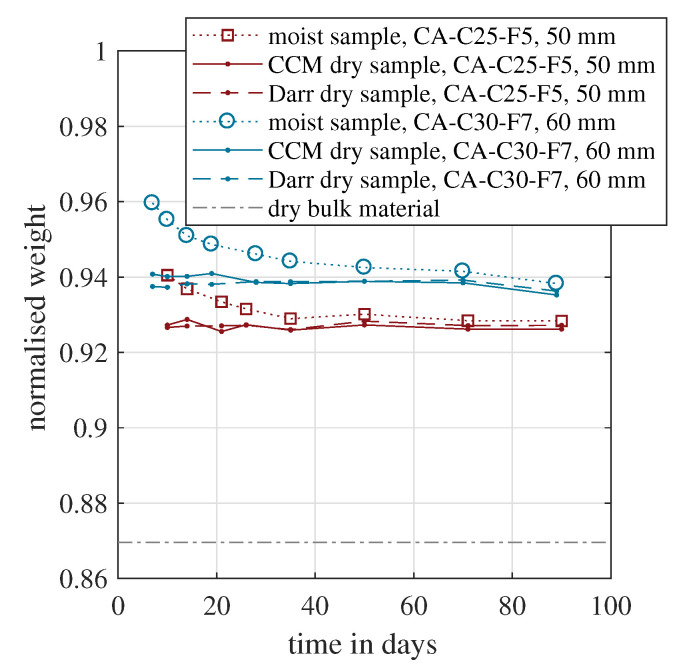
Evolution of the normalised dry weight over time measured for two samples with a calcium sulphate based binder.

**Figure 7 materials-15-08422-f007:**
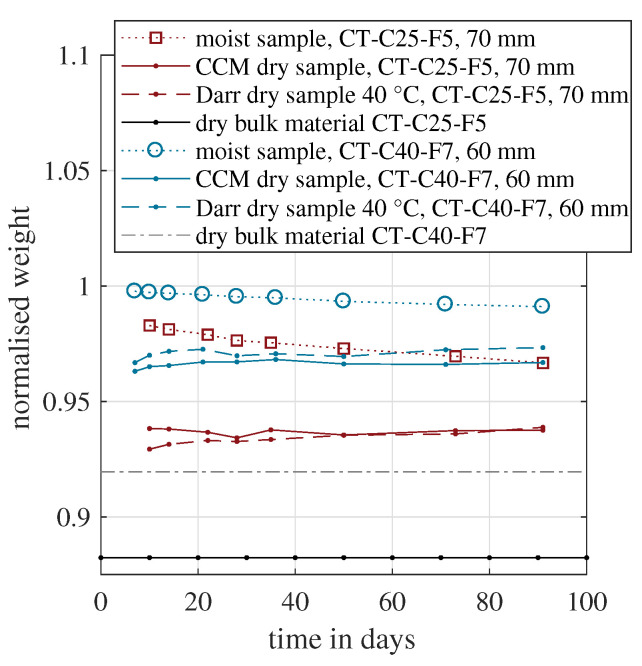
Evolution of the normalised dry weight over time measured for two samples with a cement-based binder.

**Figure 8 materials-15-08422-f008:**
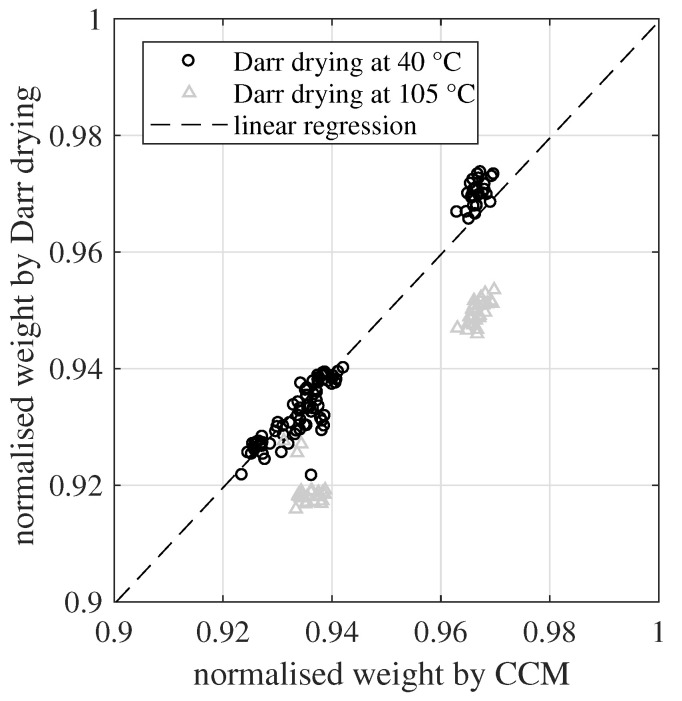
Linear regression of the normalised weight based on the CCM and Darr drying (at 40 °C). The regression comprises only a slope coefficient, excluding a y-axis intercept.

**Table 1 materials-15-08422-t001:** Classification of the used cement-based and calcium sulphate based screed types.

Binder	Type	Compressive Strength	Bending Tensile Strength	Water Demand	Aggregate Size	Label
Cement (CT)	Screed	25 N mm${}^{-2}$	5 N mm${}^{-2}$	0.1333 L kg${}^{-1}$	0–5 mm	CT-C25-F5
Cement (CT)	Rapid screed	40 N mm${}^{-2}$	7 N mm${}^{-2}$	0.0875 L kg${}^{-1}$	0–4 mm	CT-C40-F7
Calcium sulphate (CA)	Floating screed	25 N mm${}^{-2}$	5 N mm${}^{-2}$	0.15 L kg${}^{-1}$	0–4 mm	CA-C25-F5
Calcium sulphate (CA)	Floating screed	30 N mm${}^{-2}$	7 N mm${}^{-2}$	0.15 L kg${}^{-1}$	0–2 mm	CA-C30-F7

Product name of the used screeds in the order of Table 1: maxit plan 425, maxit plan 435, maxit plan 470 and maxit plan 490.

## Data Availability

Upon request, all moisture data will be shared and sent by email.

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
