# Peer review of "Comparison of the Calcium Carbide Method and Darr Drying to Quantify the Amount of Chemically Bound Water in Early Age Concrete"

_materials, 2022, doi:10.3390/ma15238422_

Round 1

Reviewer 1 Report

Summary

Hydration assessment is of great interest for various building materials. To assess hydration, the time-consuming Darr drying method is often used, which has several systematic measurement uncertainties, such as choosing the correct drying temperature. This study presents and evaluates CCM as an alternative method. The CCM directly measures the amount of water evaporated and can therefore track hydration. This method gives state-of-the-art results, can be used in the field, and covers all types of material and binder formulations.

Comments on the general concept

Formulation of the problem.

The authors of the article aim to compare the method of calcium carbide and the drying of Darra to quantify the amount of chemically bound water in early-age concrete. This study discusses the material properties of four samples, including two different binders. In this study, two cement-based materials and two calcium sulfate-based materials were investigated. Sample preparation, mixing and storage in a climate chamber are described. The procedure for measuring CCM to determine the moisture content of the material is presented in detail. The results of the study are presented clearly, contribute to the scientific field of study, are fully and correctly formulated. The conclusions are supported by the analysis of the results, provide answers to research questions.

Remarks

1. The motivation section (1 paragraph) provides arguments about the global anthropogenic mass and global CO2 emissions. This section mentions the formation of waste during the demolition of buildings, as well as the increase in transport loads. References are given to works [1-12] concerning these issues. This element of the review is not entirely appropriate for this article. Because next comes the analysis of hydration processes described by various authors. Theoretical issues of determining the amount of physically and chemically bound water are presented. This is the main and strong part of the work.

2. In this study, two cement-based materials and two calcium sulfate-based materials were investigated. Their markings are given in the text (CA-C25-F5 and CA-C30-F7, CT-C25-F5 and CT-C40-F7). In the materials section, it is necessary to comment on such marking.

The shortcomings noted are not of a fundamental nature. They do not affect the scientific content of the article.

Conclusion

The scientific article is quite informative for specialists and is of interest to researchers involved in the problem of hydration of cement materials. The proposed CMC method can be used in field and laboratory conditions.

The article can be recommended for publication.

Author Response

  1. The motivation section (1 paragraph) provides arguments about the global anthropogenic mass and global CO2 emissions. This section mentions the formation of waste during the demolition of buildings, as well as the increase in transport loads. References are given to works [1-12] concerning these issues. This element of the review is not entirely appropriate for this article. Because next comes the analysis of hydration processes described by various authors. Theoretical issues of determining the amount of physically and chemically bound water are presented. This is the main and strong part of the work.

Answer: Dear referee, indeed, the introduction starts with a very global view on the construction sector, which is of course not the main topic. Nevertheless, we emphasize several times the need of a method which works in the field and for different types of binders. They are required in the future for a CO2 reduced construction industry and more sustainable building materials. To strength this point, we added in the conclusion:

“This makes the method highly suitable to monitor the DoH of CO2-reduced binders, new mixtures, or recycled materials. In case of unexpected ambient conditions such as heat and drought periods, this in situ methods which delivers prompt results enables a dense DoH monitoring if required.”

  1. In this study, two cement-based materials and two calcium sulfate-based materials were investigated. Their markings are given in the text (CA-C25-F5 and CA-C30-F7, CT-C25-F5 and CT-C40-F7). In the materials section, it is necessary to comment on such marking.

Answer: Indeed, and we already did. It is the last column in Table 1 in section 3.1. In this column with the title “label” the markings are given. Unfortunately, you are not able to see this in the PDF. Although we use the official template, it cuts off parts of the table. We guess in the final layout; the editor will fix this. 

Reviewer 2 Report

This manuscript was well written titled as “Comparison of the calcium-carbide-method and Darr drying to quantify the amount of chemically bound water in early age concrete”. There are some questions about the contents, please carefully consider. Thanks.

1. In abstract, “Furthermore the determination of the “correct” drying temperature is still an open question.”, there was little results about this in the content, please consider carefully.

2. There was little result about the physisorption in Keywords.

3. The methods about Darr drying and CCM were not very clear for the reader although they were mentioned in 2.3 Section. Please make them clearer.

4. Please point out the most remarkable features for the Darr drying and CCM in Section 2.

5. What’s the purpose for the different height of the samples, “50mm, 60mm, and 70mm”, the direct results were not so obviously.

6. How to define the “early age”? It has a relatively clear definition for the concrete, is it different for the cement mortar?

7. In Conclusion Section, it says “that the hydration is already completed in the first week after concreting” in line 452, I don’t agree with that. Please carefully demonstrate this item.

Author Response

  1. In abstract, “Furthermore the determination of the “correct” drying temperature is still an open question.”, there was little results about this in the content, please consider carefully.

Answer: We addressed this topic in Figure 4 and 5 and in the corresponding paragraphs. Due to the release of chemically bound water at 105 °C, we excluded these results for the further analyses. However, we do see your point that we do not suggest or evaluate the “Correct” drying temperature. To avoid confusion, we delete this sentence from the abstract as it is not the focus of the paper.

  1. There was little result about the physisorption in Keywords.

Answer: To account more for physisorption, we added the following

“The difference between both is the physiosorbed mass of moisture $m_{H_2O}$”

Furthermore, we replaced “evaporable” several times with “physiosorbed”.

Also in Figure 1, physisorption was already shown and its influence on the entire moisture balance. If you believe it is still too less or misleading, we can of course delete this keyword.

  1. The methods about Darr drying and CCM were not very clear for the reader although they were mentioned in 2.3 Section. Please make them clearer.

Answer: We believe that both Darr and CCM method is explained in detail. But we agree that the information was loosely distributed between the fairly long chapters. Therefore, we restructured the entire paper and added for instance the chapter state of the art, experimental approach or testing procedure. This should give more guidance to the reader.
A detailed description for CCM can now be found in section 5 and 6, Darr drying is closely described in section 6.

  1. 4. Please point out the most remarkable features for the Darr drying and CCM in Section 2.

From our point of view these points are met in section 3. The second paragraph deals with the Darr Method, the last paragraph considers CCM. We hope that the restructuring makes it easier to find and follow the statements.

  1. What’s the purpose for the different height of the samples, “50mm, 60mm, and 70mm”, the direct results were not so obviously.

Answer: This study was part of a bigger project. There, water damage due to pipe leakage in subfloors were investigated with Radar and Neutron probe. Three different subfloor thicknesses were produced, which represents “normal” subfloor constructions. Therefore, the thickness variation was not designed for this study. Nevertheless, three samples of the same binder enable a better comparison and consistency check. But indeed, there are no direct result mentioned in the paper regarding the thickness variation in respect to the chemisorption process, because there is no indication in our data that the thickness would have any influence. To account for this, we included in the new sub-section 7.3 the following sentence:

“Furthermore, looking on the residuals of the regression, there is no indication that the different sample thicknesses have any influence on the determination of the DoH.”

  1. How to define the “early age”? It has a relatively clear definition for the concrete, is it different for the cement mortar?

Answer: In Germany, the difference of concrete, screed, and mortar is just the size of the aggregates. It does not say anything about the binder. Thus, regarding the cement matrix, we consider “early age” as the same for all of them. To make this clear, we add the following sentence in the paper:

“The binder of the screed is identical to concrete, just the maximum aggregate size is 8 mm, which is higher for concrete.”

  1. In Conclusion Section, it says “that the hydration is already completed in the first week after concreting” in line 452, I don’t agree with that. Please carefully demonstrate this item.

Answer: Dear referee, I do see your point. I believe it’s a question of the wording. If hydration describes solely the process of chemisorption of water, it would be finished after 7 days within the measurement accuracy of the used methods. Nevertheless, if hydration describes the entire chemical reaction and phase transition from silica fume, alite, portlandite, ettringite towards calcium silica hydrates, this process takes decades or might run to infinite time. However, based on the shown results, it seems that this part of hydration/ microstructural transition does not consume further water.

In conclusion, we agree that hydration is a misleading term in this context. We made the following chances to point 4 in the conclusions:

We replace the word hydration by chemisorption.

“The quantification of chemically bound water with the Darr and the CCM method shows for both methods that no significant chemisorption of free water occurs between measurement day 7 and 90 for the calcium-sulphate-based and the cement-based materials.”

We deleted the following sentence:

“In fact, the presented material moisture measurements indicate that the hydration is already completed in the first week after concreting”

Reviewer 3 Report

Revise the abstract. It does not state the objective, methods, and results in an appropriate manner.

Captions of tables should be above the table.

Sectioning is confusing. It should be done again. Experimental/Materials/Testing methods

Fig.3 is above the section in which it is referred. Please check.

Statistical analysis based on regression is needed to compare the Darr method.
Conclusions are short and does not comprehend the work.

Was chemically bound water was determined using analytical characterizations to corroborate with Darr and Calcium Carbide method? if not? why not.

Why is Darr method so intriguing to authors? What is the motive behind the study. It shall be explained in the introduction section.

Overall. The paper is written in a complex manner. The author should make an effort to rewrite in an organized and simple manner. Editing and proofread services can be availed.

Author Response

  1. Revise the abstract. It does not state the objective, methods, and results in an appropriate manner.

Answer: Thank you for the comment. We agree that the abstract’s structure was not that obvious and clear, which is why it was revised. While the objective was, in our point of view, already given in the first part, the second part was extended by additional sentences about the methods and the achieved results.  

  1. Captions of tables should be above the table.

Answer: Thanks, we changed this.

  1. Sectioning is confusing. It should be done again. Experimental/Materials/Testing methods

Answer: Thank you for this suggestion. We restructured and split the long Motivation and Theory chapters into: - a short Motivation – Theory to chemisorption – State of the art – Materials – Experimental approach – Test procedure

We hope this leads the reader better and increase the clarity.

  1. Fig.3 is above the section in which it is referred. Please check.

Answer: We used the official MDPI template and placed the figure below the paragraph where it is mentioned the first time, in accordance with the guidelines of MDPI. We do not know why it shifts the figure to the top. We would like to stay with the guidelines and expect that the editor will move this at the correct position.

  1. Statistical analysis based on regression is needed to compare the Darr method.
    Conclusions are short and does not comprehend the work.

Answer: We added an additional sub-section “Comparison of Darr drying and CCM applied to quantify chemisorbed water” and placed it directly before the Conclusion. There, we show the linear regression. We hope that this concludes the findings more comprehensively and gives a direct link to the conclusion.

  1. Was chemically bound water was determined using analytical characterizations to corroborate with Darr and Calcium Carbide method? if not? why not.

Answer: We conducted Neutron Probe and NMR relaxometry measurements as well. In theory, both methods are claimed to be able to distinguish between free and chemically bound water. Nevertheless, the moisture variations were small. Both methods were able to detect a variation in the total material moisture content, but a derivation of the chemically bound water not measurable within the given signal noise. Therefore, we decided to omit a comparison as it does not contain any usable information/ correlation.

The only analytical characterisation known to us is the Karl Fischer titration. We do not have such a device in our department. Although this method would be a very interesting validation due to its high precision, the Darr method is considered as the “golden” reference in civil engineering (although there is even not a clear definition of the drying conditions). Furthermore, our focus was on methods which are “common” in civil engineering/ on the construction side. That’s why we show only Darr and CCM data.

  1. Why is Darr method so intriguing to authors? What is the motive behind the study. It shall be explained in the introduction section.

Answer: As mentioned in the second paragraph of section 3, Darr drying is the most common method to determine the amount of free water in a material sample. However, it is also controversially discussed due to the disadvantages mentioned in the same paragraph. The last paragraph of section 3 then leads to CCM as a promising alternative, which is investigated in this study.

An overall motivation is now added in the first section “Motivation”:

“New binders, new mixtures, use of recycled materials or aggregates, they all represent very promising approaches. Creating confidence in these new materials requires reliable in situ characterisation. In construction, one crucial parameter is the degree of hydration (DoH). For instance, on this parameter the current and final compressive strength are estimated or concrete post-treatment actions are scheduled and optimised. Therefore, this study discusses an approach for quantifying the DoH which works in the field as well.”

  1. Overall. The paper is written in a complex manner. The author should make an effort to rewrite in an organized and simple manner. Editing and proofread services can be availed.

Answer: We contacted an expert for proof-reading. This person also noted that several information is only loosely connected between the Motivation and Theory chapter which reduces the overview. Based on this feedback and your suggestion in Question 2, the paper was restructured. We hope that this new structure makes it easier to flow the plot.

Round 2

Reviewer 2 Report

All my doubts were answered well.

Reviewer 3 Report

The paper has been improved considerably.